# Functional exploration of the glycoside hydrolase family GH113

**Marie Couturier**[1]*, **Mélanie Touvrey-Loiodice**[1], **Nicolas Terrapon**[2], **Elodie Drula**[2,3], **Laurine Buon**[1], **Christine Chirat**[4], **Bernard Henrissat**[5,6], **William Helbert**[1]

**1** CERMAV, Univ. Grenoble Alpes, CNRS, Grenoble, France, **2** AFMB, UMR 7257 CNRS Aix-Marseille Univ., USC 1408 INRAE, Marseille, France, **3** Biodiversité et Biotechnologie Fongiques, UMR 1163, INRAE, Marseille, France, **4** Institute of Engineering (Grenoble INP), LGP2, Université Grenoble Alpes, CNRS, Grenoble, France, **5** Department of Biological Sciences, King Abdulaziz University, Jeddah, Saudi Arabia, **6** DTU Bioengineering, Technical University of Denmark, Kgs. Lyngby, Denmark

* marie.couturier@cermav.cnrs.fr

## Abstract

β-Mannans are a heterogeneous group of polysaccharides with a common main chain of β-1,4-linked mannopyranoside residues. The cleavage of β-mannan chains is catalyzed by glycoside hydrolases called β-mannanases. In the CAZy database, β-mannanases are grouped by sequence similarity in families GH5, GH26, GH113 and GH134. Family GH113 has been under-explored so far with six enzymes characterized, all from the *Firmicutes* phylum. We undertook the functional characterization of 14 enzymes from a selection of 31 covering the diversity of the family GH113. Our observations suggest that GH113 is a family with specificity towards mannans, with variations in the product profiles and modes of action. We were able to assign mannanase and mannosidase activities to four out of the five clades of the family, increasing by 200% the number of characterized GH113 members, and expanding the toolbox for fine-tuning of mannooligosaccharides.

## Introduction

β-mannans (referred to as mannans throughout this manuscript) define a heterogeneous group of polysaccharides with a common main chain of β-1,4-linked mannopyranoside residues. Mannans are found in the cell wall of algae, in plant seeds and beans, and are a major component of wood hemicelluloses (e.g. galactoglucomannan) embedding the fibril architecture of the matrix [1, 2]. Semi-crystalline mannans participate in the structural architecture of cell walls devoid of cellulose fibrils such as the algae *Porphyra sp.* [3] or *Acetabularia sp* [4–6]. In crystalline mannan, the polysaccharide structure and packing of the chains are very similar to those of cellulose [7]. Mannans can also be extracted from beans or seeds giving gums appreciated in food industry (e.g. guar gum, tara gum, carob gum; [8]). In gums, mannan chains are decorated by carbohydrate components such as galactopyranoside residues in galactomannan gums, or by non-carbohydrate groups (e.g. acetate, sulfate) depending on the botanical origin of the macromolecules. Galactomannan seed gums are already exploited as stabilizer and thickener in the food and cosmetic industries [1, 8]. However, despite their

**Data Availability Statement:** All relevant data are within the manuscript and its Supporting Information files.

**Funding:** This work was financially supported by the Institut Carnot PolyNat (ANR-17-CARN-0025-

0). Support was received from the Glyco@Alps Cross-Disciplinary Program (Grant ANR-15-IDEX-02), Labex ARCANE (Grant ANR-11-LABX-0003), and Grenoble Graduate School in Chemistry, Biology, and Health (Grant ANR-17-EURE-0003). The funders had no role in study design, data collection and analysis, decision to publish, or preparation of the manuscript.

**Competing interests:** The authors have declared that no competing interests exist.

**Abbreviations:** CAZymes, Carbohydrate-Active enZymes; DP, degree of polymerization; GH, Glycoside Hydrolase; HPAEC-PAD, High-performance anion-exchange chromatography with pulsed amperometric detection; M1, Mannose; M2, mannobiose; M3, mannotriose; M4, mannotetraose; M6, Mannohexaose.

abundance in plant cell wall mannans are under-exploited and their valorization would be key towards sustainable biorefineries. For example, glucomannans are a co-product of the pulp and paper industry, more especially when the process leads to the production of highly refined cellulose (e.g. nanocrystalline cellulose) which requires the extraction of hemicelluloses [9]. Potential applications of wood glucomannan as emulsifiers and stabilizers [10] or platform chemicals [11] have started to be evaluated. Recently, oligo-mannans were found to promote growth of some beneficial bacteria in animal and human gut microbiota, suggesting that these oligosaccharides could be used advantageously as ingredient in prebiotic formulation [12–14]. These investigations revealed that the properties of the series of oligo-mannans were correlated to their size and their degree of substitution. As examples, a degree of polymerization of 2 to 3 (DP2/DP3) had a positive effect on the growth of *Lactobacillus sp*. [15] while DP3/DP4 were more favorable for the growth of *Lactobacillus reuteri* [16]. This example illustrates the need for fine-tuning of the oligo-mannan structures, which requires appropriate enzymes with well-defined recognition properties and mode of action.

The cleavage of β-linked mannopyranose residues is catalyzed by β-mannanases (EC 3.2.1.78) found in glycoside hydrolase families: GH5, GH26, GH113, and GH134 of the Carbohydrate-Active enzyme classification (CAZy, www.cazy.org [17]). GH134 β-mannanases adopt a three -dimensional structure similar to that of lysozyme [18] and cleave the glycosidic bond with an inversion of the anomeric configuration [19]. Contrasting with family GH134, the other families of mannan-degrading enzymes belong to same structural clan GH-A. Accordingly, mannanases from families GH5, GH26 and GH113 share remote homology, notably testified by the same $(\beta/\alpha)_8$ barrel three-dimensional structure, the same catalytic amino acids (Glu/Glu, catalytic nucleophile/catalytic proton donor) and the same catalytic mechanism leading to the retention of the anomeric configuration. GH5 and GH26 are longtime known families, from which numerous members have been characterized, whereas GH113 is a more recent, less studied family. The CAZy website (October 2021) listed for the GH113 family: 1,346 protein sequences from bacterial origin, six sequences from Archaea, four sequences from viruses and nine sequences from Eukaryotic origin, reflecting the bias in taxonomical sequencing. Among Archaea and Eukaryota encoding a GH113, most of the organisms seem to be from aquatic environment. To date, six GH113 enzymes have been biochemically characterized, namely from *Alicyclobacillus acidocaldarius* TC-12-31 [20], *Alicyclobacillus sp*. [21], *Amphibacillus xylanus* NBRC 15112 [22], *Bacillus sp*. N16-5 [23], *Roseburia intestinalis* L1-82 [24] and *Faecalibacterium praunsnitzii* SL3/3 (the latter was published during the course of this study [25]). All these enzymes degrade β-mannan polysaccharides and/or oligosaccharides: endo-β-mannanase activity has been identified for four of them (i.e. the enzymes cleave randomly the mannan backbone), and reducing-end exo-mannosidase activity was described for *R. intestinalis* and *F. spraunitzii* enzymes (i.e. the enzymes release mannose from the extremity of manno-oligosaccharides). The crystal structures of three GH113 enzymes have been solved, confirming their belonging to clan GH-A [20–22]. Hence, the diversity of GH113 family remains little explored with few enzymes characterized with high sequence similarity (>54% identity on the catalytic domain sequences across enzymes).

In this context, we selected a set of 31 enzymes on the basis of protein sequence diversity, aiming at unveiling the functional diversity of the under-explored GH113 family.

## Results

### Bioinformatic analysis of the GH113 family

Analysis of the GH113-containing proteins revealed that the vast majority are composed of this sole domain, with a single exception in the marine bacterium *Ardenticatena sp*. whose

GH113 bears a Carbohydrate Binding Module (CBM) from family CBM8. Binding to cellulose has been demonstrated for one CBM8 which shares only 36% identity with the CBM8 from *Ardenticatena sp.* Many other CBM8 sequences are found attached to a variety of enzymes, including GH18 enzymes which are mostly chitinases, or, interestingly, putative mannanases from the GH5_40 subfamily. Some sequences carry a C-terminal extension that careful analysis did not allow to relate to any known CBM or any other known module. Signal peptides were identified on approximately 54% of the sequences. A distance tree was built from the catalytic domain sequences of 511 enzymes belonging to GH113 family. This tree showed two large and three small clades driven mostly by the taxonomy (Figs 1 and S1). The large Clade 1 mainly encompasses sequences belonging to the Firmicutes phylum, including all enzymes characterized to date. Most of the sequences in Clade 1 are devoid of signal peptides (83%), suggesting an intra-cellular localization of the enzymes. The other clades are unexplored so far, without any characterized representatives. Clades 2, 3 and 4 are small clades gathering enzymes from various phyla: Clade 2 gathers mostly Alphaproteobacteria and Gammaproteobacteria, Clade 3 Actinobacteria and the few eukaryotic sequences, and Clade 4 other Alphaproteobacteria and a few diverse phyla including Archae. Signal peptides were predicted on 56% for Clade 2, 84% for Clade 3 and 57% for Clade 4. Finally, the large Clade 5 is dominated

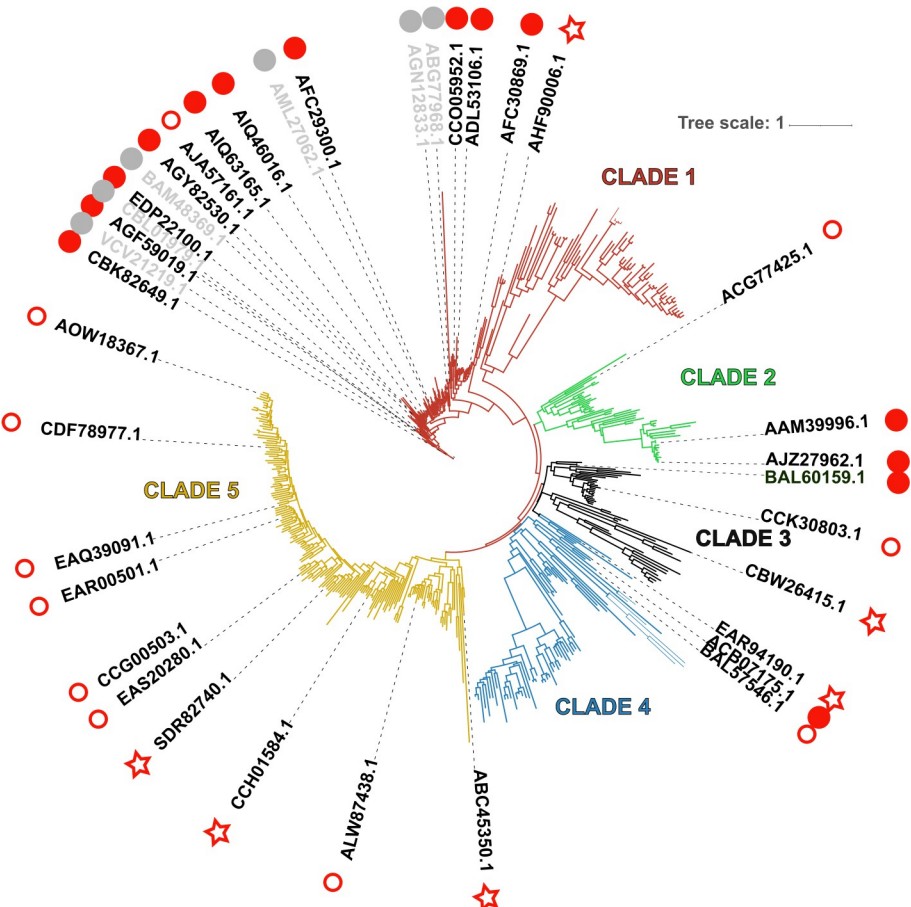

**Fig 1. Phylogenetic tree of family GH113.** Enzymes studied here are shown with red symbols: filled dots indicate that an activity was assigned to the enzyme, empty ones indicate that no activity could be demonstrated. Stars indicate that the target protein was not obtained in a soluble form. Enzymes characterized in previous studies are labeled in grey. A high-resolution phylogenetic tree is available in Supplementary Material, S1 Fig.

by Bacteroidetes species, including many strains isolated from marine environment. Signal peptides were frequently encountered in this clade (83%), suggesting an extracellular localization. The phylogenetic tree was used as a guide to select a panel of 31 enzymes, covering the diversity of the GH113 family, for biochemical investigation (Table 1 and Figs 1 and S1).

## Production of GH113 β-1,4 mannanases and substrate specificity

For this study, codon optimization for heterologous expression in *E. coli*, gene synthesis and cloning of the 31 targets were outsourced (Nzytech, Portugal). Signal peptides and other extensions were trimmed in order to conserve only the predicted catalytic domain. Twenty-five targets (81%) were successfully overexpressed in *E. coli* BL21(DE3) and soluble. The remaining six proteins were produced as insoluble inclusion body, notably the eukaryotic GH113 from *Tetrahymena thermophila* and an enzyme from the extremely halophilic bacterium *Salinibacter ruber*. The purity of enzymes was attested by a single band visible on SDS-PAGE and estimated to be >90%. The molecular weights were ranging from 35 kDa to 47 kDa, as expected. Final protein concentrations were between approximately 2 and 6 mg.mL$^{-1}$. All the enzymes were produced and screened at least twice to demonstrate reproducibility.

At first, the purified proteins were screened for activity on a set of various substrates, including several mannan polysaccharides from different biological origins (e.g. β-1,4-mannan, glucomannan, galactomannan) (S2 Fig), manno-oligosaccharides and *p*NP-β-mannopyranoside (S1 and S2 Tables). Out of the 25 soluble enzymes, 14 were found to be active on at least one mannan or manno-oligosaccharide substrate. Active enzymes were identified in all Clades but Clade 5, where no member showed even a weak activity. The seven Bacteroides proteins of Clade 5 as well as proteins from other clades for which no activity had been detected on the first set of substrates were subjected to a second screening round on a complementary set of substrates (S2 Table). Despite the diversity of polysaccharides, oligosaccharides and synthetic substrates tested, no activity was observed for any of the proteins, including all the proteins from Clade 5. Qualitative assessment of the degradation capability of the various mannans and manno-oligosaccharides by enzymes from Clades 1 to 4 is given in Table 1.

## Analysis of the mode of action for GH113 β-mannanases on debranched beta -1,4 mannan

End-products and mode of action of the enzymes were determined by size-exclusion chromatography. Endo-acting enzymes cleave the polymeric substrate into a range of oligosaccharides whose sizes decrease as the degradation proceeds. Accordingly, five enzymes of Clade 1 revealed an endo-mode of action on linear beta-1,4 mannan (Fig 2). Five other enzymes exhibited activity on a mix of linear manno-oligosaccharides, whereas no significant activity was measured on polysaccharides or on *p*NP-β-mannopyranoside. This is exemplified with EDP22100 (S3 Fig), which accumulated M2 and M1 after extended incubation with oligosaccharides. The activity of EDP22100 was further examined by HPAEC-PAD (Fig 3A) confirming the release of mannose from mannohexaose (M6). Intermediate products were detected in the shorter incubation times and quickly degraded. This analysis confirmed the exo-mannosidase activity that was also described in another *F. prausnitzii* strain during the course of our study [25]. To elucidate whether they were mannobiosidases or mannosidases, the four other enzymes with activity on oligosaccharides were similarly subjected to HPAEC-PAD analysis (Fig 3B). All of them were able to release mannose from M6, confirming that they are also exo-mannosidases.

Clade 1 endo-acting enzymes were active on all tested mannans (e.g. glucomannan, galactomannan, beta-1,4-mannan) and released mostly mannobiose (M2) and mannotriose (M3) as

**Table 1. List of enzymes studied in this work, their characteristics and qualitative activities on mannan substrates.**

| Strains | GenBank | SigP | Sol | GlcM | GalM | M | Oligo-M | Mode of action |
|---|---|---|---|---|---|---|---|---|
| **Clade 1** | | | | | | | | |
| *Opitutaceae bacterium* TAV5 | AHF90006.1 | N | - | na | na | na | na | |
| *Ruminococcus bicirculans* | CCO05952.1 | N | + | + | + | + (M2) | nd | Endo |
| *Clostridium cellulovorans* 743B | ADL53106.1 | N | + | + | + | + (M2) | nd | Endo |
| *Paenibacillus mucilaginosus* 3016 | AFC30869.1 | N | + | + | + | + (M2) | nd | Endo |
| *Paenibacillus mucilaginosus* 3016 | AFC29300.1 | N | + | + | + | + (M2) | nd | Endo |
| *Paenibacillus stellifer* | AIQ63165.1 | N | + | - | - | - | + (M2+M1) | Exo |
| *Paenibacillus sp.* FSL R7-0273 | AIQ46016.1 | N | + | - | - | - | + (M2+M1) | Exo |
| *Carnobacterium sp.* WN1359 | AGY82530.1 | N | + | - | - | - | + (M2+M1) | Exo |
| *Lactococcus lactis subsp. Lactis* | AJA57161.1 | N | + | - | - | - | - | |
| *Faecalibacterium prausnitzii* M21/2 | EDP22100.1 | N | + | - | - | (+) | + (M2+M1) | Exo |
| *Coprococcus sp.* ART55/1 | CBK82649.1 | N | + | + | + | + | nd | Endo |
| *Clostridium saccharoperbutylacetonicum* N1-4(HMT) | AGF59019.1 | N | + | - | - | - | + (M2+M1) | Exo |
| *Alicyclobacillus acidocaldarius* TC-12-31 (*Aa*Man113A)* | ABG77968.1 | | | | | | | Endo* |
| *Alicyclobacillus sp.* (Man113A)* | AGN12833.1 | | | | | | | Endo* |
| *Amphibacillus xylanus* NBRC 15112 (*Ax*Man113A)* | BAM48369.1 | | | | | | | Endo* |
| *Bacillus sp.* N16-5 (Man113A)* | AML27062.1 | | | | | | | Endo* |
| *Faecalibacterium prausnitzii* SL3/3 | CBL01979.1 | | | | | | | Exo* |
| *Roseburia intestinalis* L1-82 (*Ri*GH113)* | VCV21219.1 | | | | | | | Exo* |
| **Clade 2** | | | | | | | | |
| *Phenylobacterium zucineum* HLK1 | ACG77425.1 | N | + | - | - | - | - | |
| *Xanthomonas campestris str.* ATCC 33913 | AAM39996.1 | Y | + | + | + | + (M3+M4) | nd | Endo |
| *Xanthomonas citri subsp. citri* | AJZ27962.1 | Y | + | + | + | + (M3+M4) | nd | Endo |
| **Clade 3** | | | | | | | | |
| *Candidatus Acetothermus autotrophicum* | BAL60159.1 | N | + | - | - | + (M3+M4) | nd | Endo |
| *Streptomyces davawensis* JCM 4913 | CCK30803.1 | N | + | - | - | - | - | |
| *Tetrahymena thermophila* SB210 | EAR94190.1 | Y | - | na | na | na | na | |
| *Halobacteriovorax marinus* SJ | CBW26415.1 | N | - | na | na | na | na | |
| **Clade 4** | | | | | | | | |
| *Candidatus Korarchaeum cryptofilum* OPF8 | ACB07175.1 | Y | + | + | + | + (M3+M4) | nd | Endo |
| *uncultured Acetothermia bacterium* | BAL57546.1 | Y | + | - | - | - | - | |
| **Clade 5** | | | | | | | | |
| *Salinibacter ruber* DSM 13855 | ABC45350.1 | N | - | na | na | na | na | |
| *Hymenobacter sp.* DG5B | ALW87438.1 | N | + | - | - | - | - | |
| *Gramella sp.* MAR_2010_102 | SDR82740.1 | N | - | na | na | na | na | |
| *uncultured Flavobacteriia bacterium* | CCG00503.1 | Y | + | - | - | - | - | |
| *Maribacter sp.* HTCC2170 | EAR00501.1 | Y | + | - | - | - | - | |
| *Dokdonia sp.* MED134 | EAQ39091.1 | Y | + | - | - | - | - | |
| *Flavobacteria bacterium* BBFL7 | EAS20280.1 | Y | + | - | - | - | - | |
| *Polaribacter vadi* | AOW18367.1 | Y | + | - | - | - | - | |
| *Formosa agariphila* KMM 3901 | CDF78977.1 | Y | + | - | - | - | - | |
| *Fibrella aestuarina* BUZ 2 | CCH01584.1 | N | - | na | na | na | na | |

Main end-products are given into brackets for linear polysaccharides. SigP: Signal peptide (Y for predicted; N otherwise). Sol: Expression of soluble proteins. Substrates: + for degradation of the substrate; (+) for a weak degradation and—for no degradation. nd: not detected. na: not applicable

*: results from previous works. Underlined GenBank accession numbers highlight sequences for which the 3D structure was solved.

(GlcM: Glucomannan, M: β (1,4) mannan, GalM: galactomannan, OligoM: Oligo-mannans).

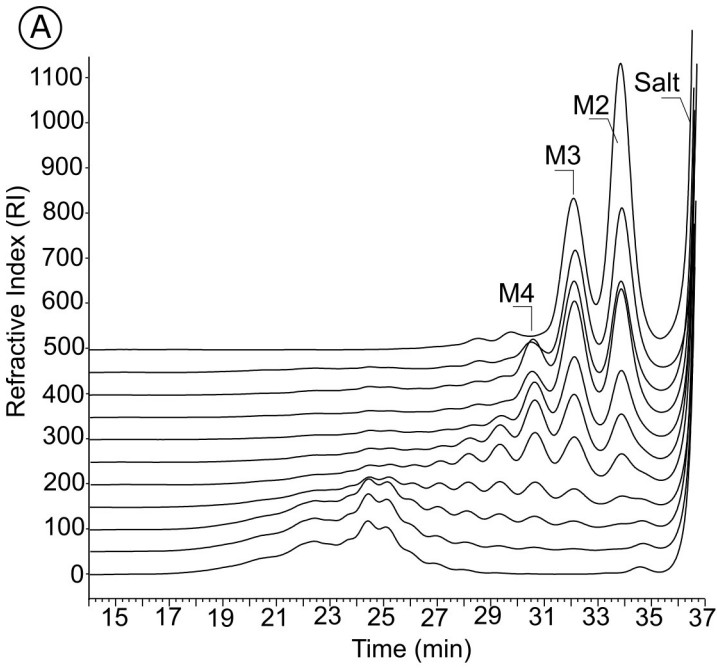

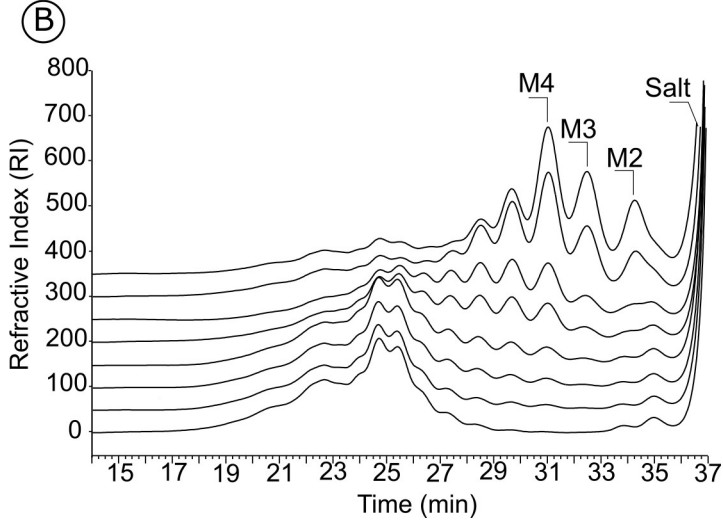

**Fig 2. Analysis of the endo mode of action of two GH113 representatives.** The degradation of debranched β 1,4 mannan by A: CCO05952 (Clade 1) and B: AJZ27962 (Clade 2) was monitored using size-exclusion chromatography. M2: mannobiose, M3: mannotriose, M4: mannotetraose.

main end-products as shown with CCO05952 in Fig 3A. Four enzymes from Clades 2, 3 and 4 were also found to be active on β-mannan polysaccharides. Degradation kinetics monitored by chromatography revealed they were all endo-acting enzymes, releasing a range of oligosaccharides over the course of the reaction. mannotriose (M3) and mannotetraose (M4) were the main end products, as illustrated with the enzyme AJZ27962 on linear mannan (Fig 3B). No transglycosylation activity was observed with any of the tested enzymes in our experimental conditions. Specific activities were determined for one representative from Clades 1 and 2, CCO05952 and AJZ27962, respectively, showing that Clade 1 CCO05952 exhibits a similar

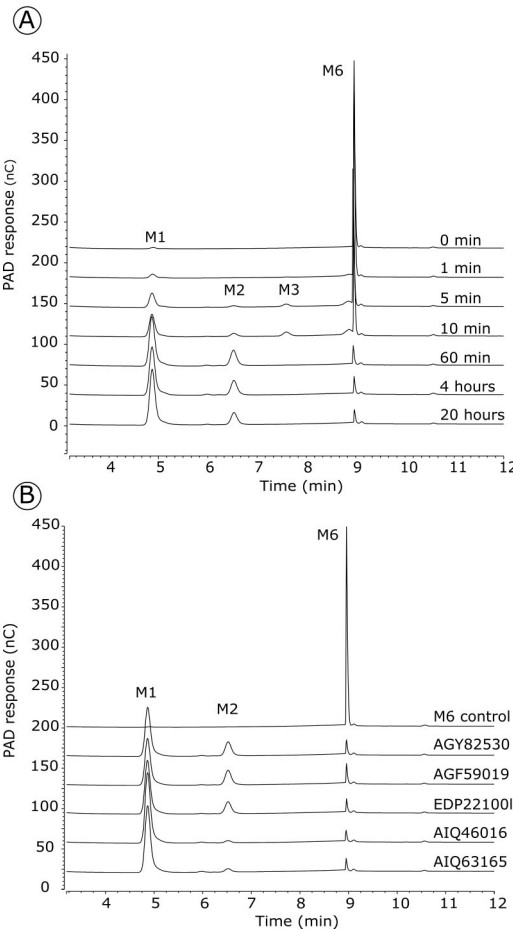

**Fig 3. Analysis of the exo mode of action of five GH113 Clade 1 representatives.** A: The degradation of mannohexaose (M6) by EDP22100 was monitored by HPAEC-PAD. B: Analysis of end-products released by AGY82530, AGF59019, AIQ46016 and AIQ63165 after overnight incubation with M6. M1: Mannose, M2: mannobiose, M3: mannotriose, M6: mannohexaose.

activity on various mannans, whereas Clade 2 AJZ27962 has highest activity on glucomannan and decreasing activity with increasing galactose substitutions (Table 2).

## Discussion

### Substrate specificity in the GH113 family

Activity on β-1,4-linked mannose-containing substrates was demonstrated for enzymes grouped in Clades 1 to 4 of the GH113 family. The specificity and the mode of action of these

**Table 2. Specific activity of representatives of M1+ M2 producing enzymes (AJZ27962.1, Clade 1) and M3+M4 producing enzymes (CCO05952.1, Clade 2) towards mannans.**

|  | Specific activity (U/mg) | |
| --- | --- | --- |
|  | **AJZ27962.1** | **CCO05952.1** |
| Konjac glucomannan (0% Gal, mannose:glucose = 60:40) | 9.39 +/-0.51 | 3.00 +/-0.30 |
| Carob linear β-1,4 mannan (0% Gal) | 4.73 +/-0.64 | 3.35 +/-0.0 |
| Carob galactomannan (22% Gal) | 4.14 +/-0.45 | 3.04 +/-0.0 |
| Guar gum (50% Gal) | 0.89 +/-0.34 | 2.79 +/-0.59 |

enzymes were carefully determined using a variety of mannan substrates. These analyses revealed different degradation profiles correlated to the mode of action and probably to the active site topology of the enzymes. Five enzymes of Clade 1 were found to be endo-acting enzymes producing mainly M2 and M3 end-products similarly to most of the other previously characterized enzymes grouped in this Clade [20–23]. Five enzymes of Clade 1 had exo-mannosidase activity on oligomannans. All the enzymes grouped in Clade 1 were inactive against *p*NP-β-mannopyranoside, suggesting that productive binding requires at least two sub-sites. Inspection of sequence alignment showed that amino acids involved in the catalytic machinery and interaction with the substrates deduced from crystallographic studies [21, 22] are well conserved in this clade (S4 Fig). In addition, it did not reveal significant differences between endo- and exo-acting enzymes such as, for example, additional loop that could modify the active site topology or absence of specific amino acids that could be involved specially in substrate interactions. This probably explains that exo- and endo-acting enzymes do not group as independent branches in the phylogenetic tree.

To the best of our knowledge, no member of other Clades than the Firmicutes Clade 1 had been studied before the present work. Four out of the seven enzymes selected here among Clades 2, 3 and 4 have been identified as endo-acting mannanases. They show catalytic characteristics different from what is observed for Clade 1. Notably, all of the active enzymes from these three clades exhibit a mode of action leading to the production of longer oligosaccharides than enzymes from Clade 1, mannotriose (M3) and mannotetraose (M4) being the main end-products. One can hypothesize that the active site should be able to accommodate longer substrates and probably presents additional sub-sites to those described in crystallographic studies of Clade 1 mannanases. Accordingly, we notice that enzymes grouped in Clade 1 are generally devoid of signal peptide, whereas enzymes of Clades 2, 3 and 4 more frequently encode one. Therefore, one can hypothesize that the former are in charge of processing internalized mannan oligomers, whereas the latter might target longer polysaccharides outside of the cell. Accordingly, the studies of *R. intestinalis Ri*113A [24] and *Bacillus sp*. N16-5 [23] revealed that the GH113-encoding genes were co-localized in the genomes with genes encoding other enzymes involved in the breakdown of glucomannan, galactomannan and galactoglucomannan. Among these enzymes, endo-acting mannanases classified in GH5 or GH26 were identified as the first degraders of the polysaccharide chains. However, this is not supported by an analysis of CAZyme content for Clade 1 species, which revealed that a majority of these species do not encode any other mannan-degrading enzyme from known CAZy family (S5 Fig).

## GH113 of unknown function

Beyond the assignment of mannanase and mannobiosidase activities of Clades 1 to 4, we were not able to identify the specific substrate of selected enzymes grouped in the large Clade 5. Sequence alignments showed that amino acids involved in the substrate recognition as well as those required for the catalytic machinery are well conserved (S4 Fig). Although disappointing, our observations confirm the importance of biochemical characterization to ascribe function of putative enzymes grouped in CAZy families. Challenging bioinformatics predictions with experimental data allows refining annotation methods and highlights the limitations of functional prediction and the attribution of a target substrate. Clade 5 mainly groups enzymes found in Bacteroidetes species living in marine environment. Therefore, the preferred substrate of these enzymes could be a poly- or oligosaccharide occurring in a marine organism but absent of our collection of substrates. This illustrates that functional characterization of CAZymes is often limited by the availability of substrates. Bacteroidetes species are known for displaying a large arsenal of Polysaccharide Utilization Loci (PUL, http://www.cazy.org/

PULDB/, [26]), genomic organizations where genes encoding all the activities leading to the complete degradation of a specific polysaccharide (import, cleavage, regulation) are co-localized and co-regulated. Therefore, the occurrence of a gene in a PUL may help to infer its functional role in the catabolism pathway. In the case of the Bacteroidetes GH113 (Clade 5) there is a small number of genes predicted in a PUL compared to the size of the family (26/661, 4%). Moreover, these PULs contain none or a single additional degradative CAZyme (only three *Cyclobacterium* species having two GH144-encoding genes, usually involved in β-1,2 glucan breakdown). This suggests that these enzymes are unlikely to participate in a complex polysaccharide uptake. They might instead be used for the construction and/or remodeling of biofilm or an energy storage polysaccharide in these bacteria, as suggested by their presence on the genome close to Glycosyltransferase (GT) genes of family GT2, encoding enzymes involved in polysaccharide synthesis. Accordingly, we observed that among the large diversity of specificities in family GT2, some of the GT2 located in GH113-containing PULs exhibit similarity with mannosyltransferases.

## Conclusion

The rational exploration of the diversity of GH113 family revealed that this family is probably a family targeting specifically mannans. The phylogenetic analysis suggested that all six members characterized so far belong to a unique clade mostly restricted to the Firmicutes phylum for which we further characterized six additional members. We further investigated three other clades without characterized members so far, that cover a broader taxonomic diversity, and we demonstrated the mannanase activity of GH113 from two Proteobacteria, an unclassified Bacteria and an Archeae. In a large fifth clade, specific of the Bacteroidetes phylum, despite the conservation of key amino acids, we were unable to ascribe a function to these proteins or even confirm that they are glycoside hydrolases.

This study increases by 200% the number of characterized GH113 and, interestingly, we observed different modes of action (e.g. endo-/exo) and various modalities of substrate recognition leading to different end-products. The set of enzymes studied herein complete the enzymes toolbox for the preparation of series oligo-mannans that may find some application, for example to produce prebiotics.

## Materials and methods

### Bioinformatics analyses and sequence selection

Protein accessions of non-fragmentary GH113 enzymes were extracted from the CAZy database (May 2021) and used to retrieve the corresponding amino-acid sequences from the NCBI database. The amino acid sequences were trimmed to isolate the GH113 catalytic domain. To reduce the initial set of 1319 sequences, we performed a CD-HIT [27] with a sequence identity threshold of 95%. The resulting 511 sequences were aligned using MAFFT tool (version 7.453, [28]) following the accuracy-oriented method with the option maxiterate of 1000. TrimAl v1.2 [29] was used to automatically remove poorly aligned regions from the alignment with the automated option. The alignment was used to compute a distance matrix based on maximum likelihood distances [30]. The resulting matrix distance was then used to construct a phylogenetic tree using FastME [31]. Peptide signals were predicted using Phobius [32].

### Synthesis of GH113 encoding genes

The selected GH113 sequences are listed in Table 1. After removal of the signal peptide sequence when present and codon optimization for expression in *Escherichia coli*, the

sequences were synthesized and inserted in pHTP1 expression vector in frame with a (His)$_6$-tag located at the N-terminal end of the proteins (NZyTech, Portugal).

## Heterologous expression and purification of recombinant GH113

Approximately 20 ng of each plasmid were used to transform *E. coli* BL21(DE3) competent cells. Resulting clones harboring the recombinant expression plasmids were grown in 5 ml LB precultures supplemented with 30 μg/mL kanamycin in a shaking incubator at 180 rpm and 37˚C overnight. Cultures were subsequently carried out in 50 ml of NZY auto-induction LB media (NZYtech) medium supplemented with 30 μg/mL kanamycin, with 1/100 v/v overnight precultures and further incubated at 25˚C for 30 hrs. Cultures were stopped by centrifugation at 4000*g* for 5 min. The bacterial pellet was resuspended in buffer A (50 mm Tris pH7.8, 300 mM NaCl, 10 mM Imidazole) with 50 μg/ml lysozyme and stored at -80˚C overnight. After thawing, resuspended cells were incubated with DNAse at 10˚C for 15 min. Insoluble fractions were removed by centrifugation at 40,000*g* for 30 min at 4˚C. The proteins were purified by affinity chromatography using a nickel agarose affinity resin (Ni-NTA resin, Qiagen) loaded on poly-prep® chromatography columns (Bio-Rad), as described in [33]. The resin was equilibrated with buffer A and the His$_6$-tagged recombinant enzymes were eluted with buffer A containing 300 mM imidazole. The purity of the fractions was estimated by 10% SDS-PAGE analysis. Before biochemical characterization, selected proteins were further purified on a gel permeation ENRich650 column (Bio-Rad) and eluted in 20 mM Tris-HCl pH 7.8, 100 mM NaCl. The purity of the fractions was assessed using 10% SDS-PAGE analysis.

## Enzyme activity screening

**Synthetic substrate assay.** The list of *p*-nitrophenyl substrates (Sigma Aldrich) used in this study is given in S2 Table. Activities toward 5 mM pNP substrates were determined by measuring the release of 4-nitrophenol in 100 mM Tris HCl pH 7.8, NaCl 50 mM, using a 200-μl reaction volume and suitably diluted enzymes. Reactions mixtures were incubated at room temperature for 2 to 16 hrs and the release of 4-nitrophenol was quantified at 405 nm.

**Polysaccharide assay.** Polysaccharide originating from plant cell wall, seeds, animal and marine environment, were screened. The list of polysaccharide substrates tested in this study is available in S1 Table. Substrates were prepared as 0.4% solutions or suspensions in the case of insoluble substrates (i.e. ivory nut mannan, carob debranched mannan). To identify GH113 enzyme substrates, 200 μl reactions containing 0.2% substrate and suitably diluted enzyme (0.4–10.9 μM per assay) in 100 mM Tris HCl pH7.8, 50 mM NaCl were incubated in 10 kDa-microfiltration plates (10 kDa, PES, Pall corporation) for 16 hrs at 25˚C under shaking. The reaction was terminated by filtration of reaction mixtures on a multiscreen HTS vacuum manifold (MSVMHTS00, Millipore) connected to a high-output vacuum pressure pump (Millipore). The filtrates were then analyzed by assaying reducing sugars using the ferricyanide assay [34] as described before [35].

**Oligosaccharide assay.** Oligosaccharides were prepared in-house from linear β-1,4 mannan. Ten milliliters of 0.4% mannan suspension were incubated with 100 μg of GH113 (AJZ27962.1) in Tris HCl 100 mM pH 7.8, NaCl 50 mM for 30 hrs at 25˚C. The reaction was terminated by boiling for 10 min. After filtration on a 0.22 μm PES membrane, the obtained oligosaccharides were stored at -20˚C until use. GH113 activities toward oligosaccharides were determined as follows: 100 μl of oligosaccharide solution were incubated with suitably diluted enzymes in Tris HCl 100 mM pH7.8, 50 mM NaCl and incubated for 16 hrs at 25˚C under shaking, as described previously. The reactions were terminated by boiling for 5 min and products were analyzed as described in the next section. For M6 degradation experiments, 100μM

M6 were incubated with the suitably diluted enzymes in 10 mM phosphate buffer pH 7.5 at 25˚C under shaking and the reaction was terminated by boiling for 10 min and filtered on 10 kDa cut-off PES membranes.

## Gel permeation chromatography

Enzymatic degradation of polysaccharides and oligosaccharides was validated by gel permeation chromatography using a Superdex peptide 10/300 (GE Healthcare) column connected to a high-performance liquid chromatography (HPLC) Ultimate 3000 system (Thermo Fisher). Injection volume was 50 μL and the elution was performed at 0.5 ml.min$^{-1}$ in 0.1 M NaCl. Oligosaccharides were detected by differential refractometry (Iota 2 differential refractive index detector, Precision Instruments). Mannose and mannooligosaccharides ranging from DP2 to DP4 were used as standards.

## HPAEC-PAD

Enzymatic degradation of M6 was followed using high-performance anion-exchange chromatography with pulsed amperometric detection on a Dionex ICS-6000 system. Hydrolysis products were loaded onto a CarboPac PA100 2 x 250 mm column coupled to a CarboPac PA100 2 x 50 mm guard column. Injection volume was 5 μl and flow rate was 0.25 ml/min. Elution was carried out from 0 to 15 min in NaOH 0.1 M with a gradient from 0 to 0.5 M sodium acetate. The elution was followed by cleaning and regeneration of the column with 10 min of NaOH 0.2 M and 1M sodium acetate, 10 min NaOH 0.2M and 10 min NaOH 0.1 M.

## Supporting information

**S1 Fig. High resolution phylogenetic tree of family GH113.** Major taxonomic groups were color-coded (tree branches and leaf labels): red for Firmicutes, blue for Alphaproteobacteria, green for Gammaproteobacteria, black for Actinobacteria and brown for Bacteroidetes. (PNG)

**S2 Fig. Characteristics of the mannans used in this study.** Symbolic representation of glycans is given with blue dots: glucose, green dots: mannose, yellow dots: galactose. (DOCX)

**S3 Fig. Degradation of linear mannooligosaccharides by a clade 1 GH113 representative.** The degradation of manno-oligosaccharides by EDP22100 was monitored using size-exclusion chromatography. M1: Mannose, M2: Mannobiose, M3: mannotriose, M4: mannotetraose. (DOCX)

**S4 Fig. Multiple alignment of selected GH113 sequences.** Conserved amino acids are highlighted with a red background, catalytic amino acids are indicated by a grey dot. Multiple alignment was generated using the MEGA Software (Tamura et al., 2021) and the figure was edited using ESPript 3 (Robert and Gouet, 2014).
Supplementary references
Tamura K, Stecher G, Kumar S. MEGA11: molecular evolutionary genetics analysis version 11. Mol Biol Evol. 2021;38:3022–3027. doi: 10.1093/molbev/msab120.
Robert X, Gouet P. Deciphering key features in protein structures with the new ENDscript server. Nucl Acids Res. 2014; 42(W1): W320-W324. doi: 10.1093/nar/gku316).
(DOCX)

**S5 Fig. Comparison of mannanase-encoding gene repertoires in Clade 1 organisms using double hierarchical clustering.** Top tree shows mannanase families or subfamilies, left tree

highlights clusters in Clade 1 fungi defined by number of mannanase-encoding genes. Abundance of the different genes within a family is represented by a colour scale from 0 (white) to the maximal number for each family (red) per species. The figure was edited using Morpheus (https://software.broadinstitute.org/morpheus/). A hierarchical clustering was performed with the option "one minus pearson correlation " on the rows. The linkage method used was average.
(DOCX)

**S1 Table. List of polysaccharides used in this study.** CM: Carboxymethyl, CWP: Cell wall polysaccharides.
(DOCX)

**S2 Table. List of synthetic substrates used in this study.**
(DOCX)

## Acknowledgments

Sophie Mathieu and Laurent Poulet are thanked for their help with plasmid transformation.

## Author Contributions

**Conceptualization:** Christine Chirat, William Helbert.

**Data curation:** Nicolas Terrapon, Elodie Drula.

**Funding acquisition:** Christine Chirat, William Helbert.

**Investigation:** Marie Couturier, Mélanie Touvrey-Loiodice, Nicolas Terrapon, Elodie Drula, Laurine Buon.

**Methodology:** Marie Couturier, Laurine Buon.

**Supervision:** Bernard Henrissat, William Helbert.

**Validation:** Marie Couturier, Mélanie Touvrey-Loiodice.

**Visualization:** Marie Couturier, Elodie Drula, William Helbert.

**Writing – original draft:** Marie Couturier, Nicolas Terrapon, William Helbert.

**Writing – review & editing:** Marie Couturier, Mélanie Touvrey-Loiodice, Nicolas Terrapon, Elodie Drula, Laurine Buon, Christine Chirat, Bernard Henrissat, William Helbert.

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
