## [Decision Letter · Decision Letter 0]

7 Mar 2022

PONE-D-22-02049Functional exploration of the Glycoside Hydrolase family GH113PLOS ONE

Dear Dr. Couturier,

Thank you for submitting your manuscript to PLOS ONE. After careful consideration, we feel that it has merit but does not fully meet PLOS ONE’s publication criteria as it currently stands. Therefore, we invite you to submit a revised version of the manuscript that addresses the points raised during the review process. In your revised manuscript please pay particular attention to the constructive comments of Reviewer 1.

We look forward to receiving your revised manuscript.

Kind regards,

Israel Silman

Academic Editor

PLOS ONE

Journal Requirements:

[This work was financially supported by the Institut Carnot PolyNat (ANR-17-CARN-0025-0). M.C., W.H., C.C. have received support from the Glyco@Alps Cross-Disciplinary Program (Grant ANR-15-IDEX-02), Labex ARCANE, and Grenoble Graduate School in Chemistry, Biology, and Health (Grant ANR-17-EURE-0003). Sophie Mathieu and Laurent Poulet are thanked for their help with plasmid transformation.]

 [The funders had no role in study design, data collection and analysis, decision to publish, or preparation of the manuscript.]

Reviewers' comments:

Reviewer's Responses to Questions

**Comments to the Author**

1. Is the manuscript technically sound, and do the data support the conclusions?

Reviewer #1: Partly

Reviewer #2: Yes

Reviewer #3: Yes

2. Has the statistical analysis been performed appropriately and rigorously? 

Reviewer #1: N/A

Reviewer #2: Yes

Reviewer #3: Yes

3. Have the authors made all data underlying the findings in their manuscript fully available?

Reviewer #1: Yes

Reviewer #2: Yes

Reviewer #3: Yes

4. Is the manuscript presented in an intelligible fashion and written in standard English?

Reviewer #1: Yes

Reviewer #2: Yes

Reviewer #3: Yes

5. Review Comments to the Author

Reviewer #1: Couturier et al have mapped the function of GH113 in representative parts of the phylogenetic tree. This is a much needed study on an underexplored family of carbohydrate active enzymes so far seen to contain beta-mannanases. The work is very comprehensive and of fine quality. Some points to be considered are listed below.

1. One obvious shortcoming of the functional characterization of the enzymes by describing the oligosaccharide product profiles is that the substrates used can contain galactose and glucose in addition to mannose. Can the precise structures of the various oligosaccharide products be informed?

2. In the abstract please inform that a fifth clade has not been functionally characterized.

3. Please mention that GH113 belongs to GH clan A

4. CBM8 is described in CAZy as a cellulose binding module. Can the authors inform more on the single GH113 exception that carries this CBM? (line 87)

5. It would be helpful for readers to show a figure with structures of the different substrates, which would be particularly relevant in connection with the results in Tables 1 and 2. This point of concern relates to the point #1 above. Note that konjac glucomannan contains glucose which might be reasonable to indicate in Table 2.

6. The author’s state that the activity assayed on the different oligo- and polysaccharide substrates was determined using “suitably diluted enzyme” solutions. The enzyme concentration is important to informed (for example as a range given the relevant section in Methods). In particular for cases where no activity was determined the used enzyme concentrations are particularly important to inform. Ideally the enzyme concentration in such cases should be up to 3-4 orders of magnitude higher than the concentrations used for enzymes where activity was measurable.

7. Please describe how the ivory nut mannan was solubilized as well as how the oligosaccharides thereof were prepared.

8. Please describe which mannan is used in Figure 2.

9. If structure determined GH113’s are part of the presentations perhaps indicate which are these (Figures S1, S2, more?).

10. The four chromatograms in Figure S2, what do they present? Which enzyme is used? (EDP22100, is hard to identify)

11. Are glucotrioses A and B defined?

12. Why are clades 3 and 4 not represented in Table 2?

13. The text lines 243-248 is not easy to understand

14. The GTs mentioned in line 50, would they have been identified – or is the produced polysaccharide identified - in a way that allows speculation on the specificity of Clade 5 to a level suggesting specificity with regard to carbohydrate contents in substrates?

15. Can purity and yields of recombinant enzymes be informed as SI?

Minor:

1. Replace tridimensional with three-dimensional

2. Make sure all references adopt consistently to the required style with regard to titles being in Sentence style (not Title Style), all information included, and Latin names in italics

3. Use “period sign”, not “comma” to indicate the decimal place

4. When listing the characterized enzymes (page 3) please apply italics only to the part of the name that the convention requires

5. Line 76, please after the two named species add “…enzymes.”

6. Please don’t start a sentence with a number, but spell the number out (and the unit if relevant) in such cases

7. Present abbreviations in alphabetical order

8. Consider in Methods to use “min” instead of “minutes” and “h” or “hrs” instead of “hours”. Use italics for “g” for gravity.

9. Line 218, should “where” actually be “were”?

Reviewer #2: I enjoyed reading this functional exploration of a poorly understood GH family. It was clearly written, the reasons for doing the experiment were clearly stated, experiments well designed and executed.

I was a bit surprised to see the high selectivity for mannan substrates and tthe mix of apparent endo and exo type reactions in the same family, but thats what discovery research is all about.

Reviewer #3: This is a very interesting and significant manuscript on the characterization of an under-explored family of glycoside hydrolases that hydrolyze mannan, a beta-mannose polymer of high industrial interest, especially in terms of biomass valorization. The authors were able to unveil the functional diversity of enzymes of this GH family by classifying the product profile (mannanase and mannosidase activities) of a set of 31 enzymes on the basis of protein sequence diversity. The manuscript is very well written and the results are sound and they increase our understanding of GH113 enzyme sand beta-mannanases in general. It is a very well performed multidisciplinary work, combining bioinformatics with biochemistry and chemical biology techniques, which in my opinion deserves publication in Plos One.

- The acronyms DP2, DP3 and DP4 (page2 ) should be defined.

- It is worth citing ACS Cent. Sci 2016, 2, 896 for the structural characterization of the lysozyme-like fold of GH134 beta-mannanase and catalytic mechanism

- May be the authors could define the difference between mannanases and mannosidases, which I think is not obvious to the general reader

6. PLOS authors have the option to publish the peer review history of their article (what does this mean?). If published, this will include your full peer review and any attached files.

Reviewer #1: No

Reviewer #2: No

Reviewer #3: No

---

## [Author Response · Author response to Decision Letter 0]

1 Apr 2022

The authors thank the reviewers for their careful reading of our manuscript and their kind and stimulating words. We have carefully taken into account all their comments and recommendations. Please find below our point to point answers. 

Reviewer #1: Couturier et al have mapped the function of GH113 in representative parts of the phylogenetic tree. This is a much needed study on an underexplored family of carbohydrate active enzymes so far seen to contain beta-mannanases. The work is very comprehensive and of fine quality. Some points to be considered are listed below.

1. One obvious shortcoming of the functional characterization of the enzymes by describing the oligosaccharide product profiles is that the substrates used can contain galactose and glucose in addition to mannose. Can the precise structures of the various oligosaccharide products be informed? 

The collection of substrates, including branched mannans, was used for the screening experiments. The end-products and, therefore, the specificity of the enzymes were determined using only linear mannans, either polysaccharide (unbranched beta 1,4 linear mannan from carob), or oligosaccharides (unbranched beta 1,4 linear mannan from carob digested with an endo-enzyme). Therefore, the structures of the produced oligo-saccharides were unambiguously linear and only composed of mannose. Size exclusion chromatography analyses with appropriate oligo-mannans standards allowed us to determine the DP of the released products. The end-products obtained after digestion of branched mannan (e.g. galactomannan) and glucomannan were not investigated.

To clarify this, we have modified Table 1 sothat major end products clearly relate with the corresponding substrate. 

2. In the abstract please inform that a fifth clade has not been functionally characterized.

We modified the sentence l.23: “We were able to assign mannanase and mannosidase activities to four out of the five clades of the family.”

3. Please mention that GH113 belongs to GH clan A

The following sentences are mentioned in the manuscript l.59-64 : “Contrasting with family GH134, the other families of mannan-degrading enzymes belong to same structural clan GH-A. Accordingly, mannanases from families GH5, GH26 and GH113 share remote homology, notably testified by the same (β/α)8 barrel tri-dimensional structure, the same catalytic amino acids (Glu/Glu, catalytic nucleophile/catalytic proton donor) and the same catalytic mechanism leading to the retention of the anomeric configuration.” However, we re-emphasized it l.79 “The crystal structures of three GH113 enzymes have been solved, confirming their belonging to clan GH-A”. 

4. CBM8 is described in CAZy as a cellulose binding module. Can the authors inform more on the single GH113 exception that carries this CBM? (line 87)

The CBM8-GH113 mentioned in the manuscript is from the marine organism Ardenticatena sp. The only CBM8 characterized to date exhibits binding to cellulose, but its sequence has only 36% identity with that of Ardenticatena. Among other CBM8, many are attached to GH18 enzymes (mostly chitinases), and some are attached to GH5_40 (putative mannanases) which suggests that the substrates targeted by CBM8 could be more diverse than just cellulose. Hence, not much information can be extrapolated from this CBM8 in terms of substrate prediction. We modified the text l. 88 as follows: “Analysis of the GH113 containing proteins revealed that the vast majority are composed of this sole domain, with a single exception in the marine bacteria Ardenticatena sp. whose GH113 bears a Carbohydrate Binding Module (CBM) from family CBM8. Binding to cellulose has been demonstrated for one CBM8 which shares only 36% identity with the CBM8 from Ardenticatena sp. Many other CBM8 sequences are found attached to a variety of enzymes, including GH18 enzymes which are mostly chitinases, or, interestingly, putative mannanases from GH5_40 subfamily.

5. It would be helpful for readers to show a figure with structures of the different substrates, which would be particularly relevant in connection with the results in Tables 1 and 2. This point of concern relates to the point #1 above. Note that konjac glucomannan contains glucose which might be reasonable to indicate in Table 2.

We are aware that the structural diversity of mannan substrates used in the study might be a source of confusion among the readers who are not familiar with this group of polysaccharides. As suggested by the reviewer, we added a supplementary figure S2 which provides an overview of the composition and structure on the three types of mannans used in our study. 

6. The author’s state that the activity assayed on the different oligo- and polysaccharide substrates was determined using “suitably diluted enzyme” solutions. The enzyme concentration is important to informed (for example as a range given the relevant section in Methods). In particular for cases where no activity was determined the used enzyme concentrations are particularly important to inform. Ideally the enzyme concentration in such cases should be up to 3-4 orders of magnitude higher than the concentrations used for enzymes where activity was measurable.

We acknowledge the importance of this information and we added the range of µg of enzyme used per assay in the text (l.320). 

7. Please describe how the ivory nut mannan was solubilized as well as how the oligosaccharides thereof were prepared.

The ivory nut mannan as well as beta 1,4 linear mannan from carob are insoluble in water and were actually prepared as suspensions and not as solutions. This was explicated in the materials and methods section (l.318) as follows: “Substrates were prepared as 0.4% solutions or suspensions in the case of insoluble substrates (e.g. ivory nut mannan, carob mannan)”. We have also indicated in Table S2 which mannans are insoluble. The oligosaccharides were prepared as described in the materials and methods section, l. 328.

8. Please describe which mannan is used in Figure 2.

We used debranched linear beta 1,4 mannan from carob in figure 2 and this was added in the section title (l.153), in the text (l.157) and in the figure legend (l. 172). 

9. If structure determined GH113’s are part of the presentations perhaps indicate which are these (Figures S1, S2, more?).

This request from the reviewer was not clear to us. We propose to indicate in Table 1 which GH113 enzyme three dimensional structures have been solved in previous studies.

10. The four chromatograms in Figure S2, what do they present? Which enzyme is used? (EDP22100, is hard to identify)

Figure S2 shows the degradation products of linear manno-oligosaccharides digested with one of the exo-mannosidases from clade 1 and analyzed by size exclusion chromatography. Because the mannose (M1) is not clearly separated from the salt peak using this method, we then switched to HPAEC analysis which is presented in Figure 3. We modified the title to make it more explicit as to which enzyme and substrates were used: “Degradation of linear mannooligosaccharides by a clade 1 GH113 exo-mannosidase representative”. 

11. Are glucotrioses A and B defined?

Thanks for pointing that out. We renamed the two glucotrioses with more informative names, respectively Glucosyl-(1→3)-β-D-Cellobiose and Cellobiosyl-(1→3)-β-D-Glucose in the supplementary Table S1. We also renamed the “Glucotetraose B” to the more informative name: “Cellotriosyl-(1→3)-β-D-Glucose”.

12. Why are clades 3 and 4 not represented in Table 2?

In this table we wanted to compare the specific activities of endo-enzymes with different end product patterns. Therefore, we chose enzymes not according to the clade they were located in but rather differentiating M1+M2 vs M3+M4 producing enzymes. Clades 2, 3 and 4 all gathers M3+M4 producing enzymes therefore only one was used. The title (l.196) was modified as follows: “Specific activity of representatives of M1+ M2 producing enzymes (Clade 1 (AJZ27962.1, Clade 1) and M3+M4 producing enzymes (CCO05952.1, Clade 2) towards mannans.”

13. The text lines 243-248 is not easy to understand

We rephrased this section (now l.256-261) by splitting it as follows: “In the case of the Bacteroidetes GH113 (Clade 5) there is a small number of genes predicted in a PUL compared to the size of the family (26/661, 4%). Moreover, these PULs contain none or a single additional degradative CAZyme (only three Cyclobacterium species having two GH144-encoding genes, usually involved in β-1,2 glucan breakdown). This suggests that these enzymes are unlikely to participate in a complex polysaccharide uptake.”

14. The GTs mentioned in line 50, would they have been identified – or is the produced polysaccharide identified - in a way that allows speculation on the specificity of Clade 5 to a level suggesting specificity with regard to carbohydrate contents in substrates?

The GT found with GH113 on PUL are classified in the large GT2 families, which gathers more than 280000 sequences, with very different substrates specificities. We observed that in many of these GH113-containing PULs there are three GT2: one with little identity to characterized GT2, and two others with some similarity to GT2 mannosyltransferases. We added a sentence in the manuscript, l.264. “Accordingly, we observed that among the large diversity of specificities in family GT2, some of the GT2 located in GH113-containing PULs exhibit similarity with mannosyltransferases.”

15. Can purity and yields of recombinant enzymes be informed as SI?

After purification, the purity of enzymes was attested by a single band visible on SDS-PAGE and estimated to be >90%. The molecular weights were ranging from 35 kDa to 47 kDa, as expected. Final protein concentrations were comprised between approximatively 2 and 6 mg.mL-1, for 50 ml of culture volume. We added these informations in the manuscript, l.134. 

Minor:

1. Replace tridimensional with three-dimensional

We have checked and corrected it throughout the manuscript. 

2. Make sure all references adopt consistently to the required style with regard to titles being in Sentence style (not Title Style), all information included, and Latin names in italics

We double-checked and edited accordingly the reference list. 

3. Use “period sign”, not “comma” to indicate the decimal place

Checked and corrected. 

4. When listing the characterized enzymes (page 3) please apply italics only to the part of the name that the convention requires

Corrected page 3 and in the Table 1. 

5. Line 76, please after the two named species add “…enzymes.”

Corrected. 

6. Please don’t start a sentence with a number, but spell the number out (and the unit if relevant) in such cases.

We have corrected it throughout the manuscript.

7. Present abbreviations in alphabetical order

We have corrected the abbreviation list to alphabetical order. 

8. Consider in Methods to use “min” instead of “minutes” and “h” or “hrs” instead of “hours”. Use italics for “g” for gravity.

Corrected. 

9. Line 218, should “where” actually be “were”?

Yes, thank you. Corrected. 

Reviewer #2: I enjoyed reading this functional exploration of a poorly understood GH family. It was clearly written, the reasons for doing the experiment were clearly stated, experiments well designed and executed.

I was a bit surprised to see the high selectivity for mannan substrates and tthe mix of apparent endo and exo type reactions in the same family, but thats what discovery research is all about.

We thank the reviewer for the nice comments on our manuscript. It is actually interesting that the GH113 family seems to exhibit very strong specificity for mannans, given that most CAZy families are polyspecific, such as the large and well-studied families GH5 and GH43. However, other families have also been shown to gather enzymes that act on a single family of substrate, such as GH14 (alpha-amylases) or GH134 (beta-mannanases). In the case of family GH11, similar as to what we see for GH113, all characterized enzymes so far have shown activity on xylan, with some enzymes having an endo (beta-xylanases) and others an exo (beta-xylosidases) mode of action. 

 

Reviewer #3: This is a very interesting and significant manuscript on the characterization of an under-explored family of glycoside hydrolases that hydrolyze mannan, a beta-mannose polymer of high industrial interest, especially in terms of biomass valorization. The authors were able to unveil the functional diversity of enzymes of this GH family by classifying the product profile (mannanase and mannosidase activities) of a set of 31 enzymes on the basis of protein sequence diversity. The manuscript is very well written and the results are sound and they increase our understanding of GH113 enzyme sand beta-mannanases in general. It is a very well performed multidisciplinary work, combining bioinformatics with biochemistry and chemical biology techniques, which in my opinion deserves publication in Plos One.

We thank the reviewer for the nice comments and careful reviewing of our work. Please find our answers to your questions below. 

- The acronyms DP2, DP3 and DP4 (page2 ) should be defined.

Thank you for identifying this point. We corrected the sentence l.50 as follows: “As examples, a degree of polymerization of 2 to 3 (DP2/DP3)”

- It is worth citing ACS Cent. Sci 2016, 2, 896 for the structural characterization of the lysozyme-like fold of GH134 beta-mannanase and catalytic mechanism

The suggested publication was cited in the text and added to the reference list. 

- May be the authors could define the difference between mannanases and mannosidases, which I think is not obvious to the general reader

We defined those terms l. 74 as follows: endo-β-mannanase activity has been identified for four of them (i.e. the enzymes cleave randomly the mannan backbone), and exo-mannosidase activity was described for R. intestinalis and F. spraunitzii enzymes (i.e. the enzymes release mannose from the extremity of manno-oligosaccharides). 

 

Modifications to the reference list: 

We identified two mistakes in the cited references: The paper by Lombard et al. in the first version of the manuscript was replaced by the newer CAZy database reference paper Drula et al. (Reference #17), and the paper by Drula et al. was replaced by the actual correct PULDB reference Terrapon et al. (Reference #26).

---

## [Decision Letter · Decision Letter 1]

5 Apr 2022

PONE-D-22-02049R1Functional exploration of the Glycoside Hydrolase family GH113PLOS ONE

Dear Dr. Couturier,

Thank you for submitting your manuscript to PLOS ONE. After careful consideration, we feel that it has merit but does not fully meet PLOS ONE’s publication criteria as it currently stands. Therefore, we invite you to submit a revised version of the manuscript that addresses the points raised during the review process.

In your revised manuscript please address the few remaining comments of Reviewer 1.

We look forward to receiving your revised manuscript.

Kind regards,

Israel Silman

Academic Editor

PLOS ONE

Journal Requirements:

Reviewers' comments:

Reviewer's Responses to Questions

**Comments to the Author**

1. If the authors have adequately addressed your comments raised in a previous round of review and you feel that this manuscript is now acceptable for publication, you may indicate that here to bypass the “Comments to the Author” section, enter your conflict of interest statement in the “Confidential to Editor” section, and submit your "Accept" recommendation.

Reviewer #1: All comments have been addressed

2. Is the manuscript technically sound, and do the data support the conclusions?

Reviewer #1: Yes

3. Has the statistical analysis been performed appropriately and rigorously? 

Reviewer #1: N/A

4. Have the authors made all data underlying the findings in their manuscript fully available?

Reviewer #1: Yes

5. Is the manuscript presented in an intelligible fashion and written in standard English?

Reviewer #1: Yes

6. Review Comments to the Author

Reviewer #1: Couturier et al have very well amended the manuscript on the function of GH113 representatives in the phylogenetic tree. A few small points to be considered are listed below.

1. Consider if there is sufficient details on the material (powder, milling, particle size – if possible) and suspension in case of insoluble polysaccharide substrates

2. Page 4, line 79. Is the attack of different enzymes possible from the non-reducing or the reducing end or both?

3. Page 5, line 90, correct to “bacterium”. Note also that lines 93-94 contain a repeated text from just before

4. Page 5, line 96, perhaps write “..from the GH5_40 subfamily.”

5. Please write throughout GenBank (not Genbank)

6. Page 7, line 140, perhaps write “..final protein concentrations were between approximately 2 etc..”

7. Page 9, line 178, replace “beta” with the Greek letter “�"

8. Page 15, line 326: would you be able to give the enzyme concentration range in molar units rather than weight?

9. I wonder if it is best to have the section “Conclusion” after Materials and Methods. Perhaps if before is possible that seems more reasonable.

7. PLOS authors have the option to publish the peer review history of their article (what does this mean?). If published, this will include your full peer review and any attached files.

Reviewer #1: No

---

## [Author Response · Author response to Decision Letter 1]

8 Apr 2022

Thank you for the reviewing of our revised manuscript. Here are the answers to the few points raised in the revisions: 

Reviewer #1: Couturier et al have very well amended the manuscript on the function of GH113 representatives in the phylogenetic tree. A few small points to be considered are listed below.

1. Consider if there is sufficient details on the material (powder, milling, particle size – if possible) and suspension in case of insoluble polysaccharide substrates

Both insoluble polysaccharides used in our study are commercial ones (available from Megazyme), and all available documentation is easily accessible through their website. We do not have any other information that would be relevant to the article. 

2. Page 4, line 79. Is the attack of different enzymes possible from the non-reducing or the reducing end or both?

Both R. intestinalis and F. prauznitii GH113 are reducing-end mannosidases and this was added in the manuscript, l. 76. 

3. Page 5, line 90, correct to “bacterium”. Note also that lines 93-94 contain a repeated text from just before

Both points corrected, thank you. 

4. Page 5, line 96, perhaps write “..from the GH5_40 subfamily.”

Corrected. 

5. Please write throughout GenBank (not Genbank)

Corrected. 

6. Page 7, line 140, perhaps write “..final protein concentrations were between approximately 2 etc..”

Corrected. 

7. Page 9, line 178, replace “beta” with the Greek letter “�"

Corrected. 

8. Page 15, line 326: would you be able to give the enzyme concentration range in molar units rather than weight?

Yes, the corresponding molar concentrations were added to the manuscript, l. 338. 

9. I wonder if it is best to have the section “Conclusion” after Materials and Methods. Perhaps if before is possible that seems more reasonable.

Yes, done.

---

## [Editor Report · Decision Letter 2]

11 Apr 2022

Functional exploration of the Glycoside Hydrolase family GH113

PONE-D-22-02049R2

Dear Dr. Couturier,

We’re pleased to inform you that your manuscript has been judged scientifically suitable for publication and will be formally accepted for publication once it meets all outstanding technical requirements.

Kind regards,

Israel Silman

Academic Editor

PLOS ONE
---

## [Editor Report · Acceptance letter]

14 Apr 2022

PONE-D-22-02049R2 

Functional exploration of the Glycoside Hydrolase family GH113 

Dear Dr. Couturier:

I'm pleased to inform you that your manuscript has been deemed suitable for publication in PLOS ONE. Congratulations! Your manuscript is now with our production department. 

Kind regards, 

on behalf of

Prof. Israel Silman 

Academic Editor

PLOS ONE